# Rabies Disease and Prophylaxis Knowledge Among Turkish Medical Students: Insights from a Cross-Sectional Study

**DOI:** 10.3390/tropicalmed10010009

**Published:** 2024-12-30

**Authors:** Vasfiye Demir Pervane, Pakize Gamze Erten Bucaktepe, Fatma Meral İnce, Dicle Demir, Simanur Koç

**Affiliations:** 1Family Medicine Department, Faculty of Medicine, Dicle University, Diyarbakir 21200, Türkiye; pagaerten@hotmail.com (P.G.E.B.);; 2Infectious Diseases and Clinical Microbiology, Selahaddin Eyyubi State Hospital, Diyarbakir 21100, Türkiye; 3Faculty of Medicine, Dicle University, Diyarbakir 21200, Türkiye

**Keywords:** rabies, vaccination, immunoglobulin, knowledge

## Abstract

Rabies is a fatal infectious disease that can be prevented with vaccination. The aim of this study was to evaluate the level of rabies knowledge among medical faculty students. This cross-sectional study included students in the medical faculty of a university hospital. The level of rabies knowledge was evaluated with a questionnaire evaluating 70 parameters. A total of 892 students participated in this study. Medical students knew that dogs and cats transmit rabies at high rates (96.9% and 87.4%, respectively) but understood less about other animals. Pregnancy (45.2%), being >65 years of age, having a chronic disease, and being immunosuppressed were indications for rabies vaccine, with rates between 31.4 and 37.4%. In total, 64.3% of respondents stated that the nearest health center should be approached to seek medical care, without first touching the wound. While indications for vaccination were correctly reported to be high after bites or scratches (74.2–94.6%), such indications were considered to be lower for other types of contact (46.2–66.6%). Indications for rabies immunoglobulin administration were correctly recognized at rates between 15.9% and 57.8%. Overall, the mean rabies knowledge level was 41.82 (Max. score 70). There was a statistically significant relationship between the total and subgroup rabies knowledge scores and class level, having taken a rabies course, a history of dog bites among the respondents or their family members, and possessing information about rabies. It was determined that the rabies knowledge levels among the students were insufficient. Having received lessons about rabies or a history of being bitten by an animal with suspected rabies were found to be important factors for increased knowledge about rabies.

## 1. Introduction

Rabies is a fatal infectious disease of the central nervous system, which causes acute, progressive, viral encephalitis caused by the rabies virus. This virus belongs to the Lyssavirus genus of the Rhabdoviridae family. Transmission of the disease is possible through close contact with the saliva of infected animals, by being bitten by an infected animal, or by licking injured skin and mucosa [1]. The most significant source of rabies infection is dogs in developing countries and wild animals in developed countries. In Türkiye, rabies is most commonly caused by dogs, followed by cats, livestock, and wild animals [2]. The World Health Organization (WHO) reported that approximately 59,000 people per year die from rabies, with most deaths occurring in low-income Asian and African countries [3]. Although the rabies vaccine was developed 139 years ago, rabies remains one of the most neglected viral zoonoses of the 21st century [4]. Promising treatment methods for rabies have been researched, but a rescue treatment remains to be discovered. After the development of rabies encephalitis, the clinical status among patients is fatal, with palliative care remaining the most common treatment [5].

Protective approaches against rabies can save lives. As dogs are responsible for nearly 99% of human rabies cases [6], the source of the disease can be controlled through the mass vaccination of dogs, as well as the management of post-exposure prophylaxis (PEP) and pre-exposure prophylaxis (PrEP) among both humans and animals [7].

While societal awareness of rabies is very important in controlling the disease, correct management and planning by healthcare personnel are also of vital importance after high-risk contact. Therefore, healthcare personnel must have sufficient, correct knowledge about the categories of rabies exposure, wound care, and the principles of rabies vaccine and immunoglobulin administration. Previous studies have shown that the knowledge, attitudes, and practices related to rabies are insufficient among healthcare personnel [8,9,10]. A lack of knowledge was observed most strongly among young and inexperienced healthcare workers [11]. Studies of university students have revealed insufficient knowledge levels of rabies. Among these students, those studying in the fields of human or animal health were observed to have higher levels of knowledge [12,13]. Although medical students have been shown to have higher levels of knowledge than students in other departments, studies among medical faculty students have revealed deficiencies in their rabies knowledge [14,15,16,17].

Medical faculty students who will be involved in the prevention of rabies must have sufficient education about rabies disease, prevention, and prophylaxis. Deficiencies in the knowledge, attitudes, and practices of these students should be identified so that education can account for these weak points. Educational interventions regarding rabies were shown to be effective among medical students [18]. The aim of this study was to measure the levels of knowledge among medical faculty students about rabies disease and prophylaxis. Determining rabies knowledge gaps among medical students will help healthcare instructors plan rabies education.

## 2. Materials and Methods

### 2.1. Study Design, Setting, and Participants

This cross-sectional study was prepared following the Strengthening the Reporting of Observational Studies in Epidemiology (STROBE) guidelines [19]. All the study procedures were performed in compliance with the ethical guidelines of the Helsinki Declaration. Before starting the study, the necessary institutional permission was obtained, and the study protocol was approved by the Non-Interventional Ethics Committee of Dicle University (decision no: 275, dated: 11 October 2023).

This study included students in the Medical Faculty of Dicle University (located in Diyarbakır province in the southeast of Türkiye). The study sample size calculated using the G*Power 3.1.9.7 software was a minimum of 580, with a priori analysis used to determine the difference between two independent means, providing a power of 0.95, effect size d = 0.3, and error probability α = 0.05 (two-tailed). However, considering possible refusals, withdrawals, and losses, we increased the number of participants by 50%. Thus, a total of 892 students were included in the study. The number of students included from each class was determined using the stratified random sampling method. A questionnaire based on the existing guidelines [2] and literature was prepared by the researchers, and data were collected between 1 November 2023 and 1 May 2024. Since the medical students in this study are receiving a Turkish education, the data form was prepared and applied in Turkish. The students selected for inclusion in the study were provided detailed information about the research, and all participants provided written informed consent for enrolment. The study inclusion criteria were defined as being a student in the Medical Faculty of Dicle University and volunteering to participate. Subjects who did not meet these criteria were not included.

### 2.2. Variables

The data form used in this study consisted of two sections. The first section included 10 questions to acquire data related to the participants’ age, gender, year of university study, possession of a domestic pet (and the routine vaccination status of that pet, if applicable), and history of bites or scratches from the pet. This section also recorded whether they or someone close to them had been bitten or scratched by an animal with suspected rabies, their level of rabies knowledge, their knowledge of rabies immunoglobulin, and whether or not they had received lessons about rabies. The second section of the questionnaire consisted of 70 items under 4 main sections and 10 subsections. These items measuring the level of rabies knowledge were prepared by the researchers with reference to the Türkiye Ministry of Health Field Guidelines for Rabies [2]. The responses to each item were “yes”, “no”, or “I don’t know”. Correct responses were scored with 1 point, while incorrect or “I don’t know” responses were given 0 points, providing a maximum possible score of 70 points.

To determine the level of rabies knowledge, the total knowledge level was calculated. Knowledge related to the 4 main sections of disease–agent, animal transmission–clinical, wound care, and vaccination–immunoglobulin was also calculated individually. The Disease–Agent Section contained 1 question about the agent of rabies disease and 6 questions about what kind of disease rabies is, with a maximum score of 7 points. The Animal Transmission–Clinical Section contained 13 questions about the risk of rabies infection from suspected contacts with certain animals, 13 questions about the need for a 10-day observation period after contact with certain animals with suspected rabies, and 5 questions about the symptoms that can be observed among animals with rabies disease, for a maximum score of 31 points. The Wound Care Section included 5 questions about wound care after an injury caused by an animal with suspected rabies, for a maximum score of 5 points. The Vaccination–Immunoglobulin Section included 6 questions about who can be administered rabies vaccination after a bite or scratch from an animal with suspected rabies, 11 questions about what form of suspected rabies contact requires vaccination, 6 questions about the individuals that require prophylactic rabies vaccination before the risk of rabies contact, and 4 questions about the indications for immunoglobulin administration according to the type of injury, for a maximum score of 27 points. All the students, including the 4th-, 5th-, and 6th-year students who had started clinical training, were separated into two groups: those who had received and not received rabies lessons. Analyses were then performed between these groups.

## 3. Data Analysis

The data obtained in the study were analyzed statistically using the IBM SPSS V26.0 (IBM Corporation, Armonk, NY, USA) and G*Power 3.1.9.7 software (Universität Düsseldorf: Psychologie-HHU). Descriptive tests were performed on the demographic data. Conformity of the data to normal distribution was assessed with the skewness and kurtosis indicators. Continuous variables were expressed as the mean ± standard deviation values since the distributions of the variables were normal (with indices between −1 and +1). Categorical variables were expressed as the number and percentage. Student’s *t*-test was used to compare the differences between two independent means, while the Welch ANOVA test was used to compare the mean values of six class groups, as the Levene test indicated a lack of homogeneity among the variances. When the comparisons found a significant result in the six class groups, a pairwise post hoc Tamhane T2 Test was used to determine from which group the significance originated since the variances were not homogeneous. To determine the relationships between knowledge scores, a Pearson correlation analysis was used. The hypotheses were two-sided, and a *p*< 0.05 value was accepted as statistically significant.

## 4. Results

### 4.1. Socio-Demographic, School, and Pet Ownership Characteristics

In total, 892 medical faculty students with a mean age of 22.03 ± 2.5 years (range: 17–42 years old) were evaluated. Domestic pet ownership was reported by 10.9% of the students, and 15.4% stated that they or a family member had a history of having been bitten by an animal with suspected rabies. When the students were grouped according to their rabies education status, 29.1% of all students, as well as 51.8% of the fourth-, fifth-, and sixth-year students who had started clinical training, reported that they had received rabies lessons, and 77.1% of students mentioned that had knowledge about rabies disease (Table 1).

### 4.2. The Animal Transmission–Clinical Section

In total, 96.9% of the medical students knew that rabies is spread by dogs, while 87.4% knew that rabies is spread by cats. There was also a high rate of incorrect answers (61.8–72.1%) regarding rabies transmission from farm animals (winged farm animals, horses, donkeys, cows, goats, and sheep) (Table 2). Notably, the 10-day quarantine period applies only to cats and dogs. Among the students, 88.6% correctly identified the 10-day quarantine requirement for dogs, while 73.5% recognized it for cats. While a 10-day quarantine is not recommended for other animals, such as farm and wild animals, a high percentage of students, ranging from 59.8% to 79.1%, incorrectly believed that a quarantine should be applied to such animals.

The students were asked what symptoms are observable among animals with rabies disease. Clinical findings of irritability and aggression (*n* = 861, 96.5%), increased saliva (*n* = 854, 95.7%), a desire to bite severely (*n* = 788, 88.3%), paralysis, and contractions (*n* = 773, 86.7%) were correctly identified at high rates, but the calmness and inertia that can be observed especially in occult rabies were not known among 81.5% of respondents (*n* = 727).

### 4.3. The Wound Care Section

The Turkish Ministry of Health Field Guidelines for Rabies recommends that the wound should first be thoroughly washed with soap and water and then cleaned with alcohol or one of the iodine antiseptics [2]. In total, 57.5% of the students stated that the wound should first be cleaned with alcohol or antiseptic solutions (Table 3). Early wound care following rabies-exposure risk is a crucial step in preventing transmission. However, 64.3% of the students indicated that the wound should be left untouched and that the nearest healthcare facility should be visited. It is crucial to avoid traumatizing a wound that has come into contact with the rabies virus and refrain from invasive procedures whenever possible. However, 34.8% of the students incorrectly suggested actions such as drawing blood from the wound.

### 4.4. Disease–Agent Section

We also presented questions regarding the characteristics of rabies (Table 4). We found that 12% (*n* = 107) of medical students did not know that the causative agent of rabies is a virus, and 8.9% (*n* = 79) did not know that rabies is a vaccine-preventable disease. Moreover, 23.1% of the students stated that rabies is a severe but non-fatal disease.

### 4.5. The Vaccination–Immunoglobulin Section

The students were asked to identify who could receive the rabies vaccine. We found that 45.2% of students did not know a vaccine could be given to pregnant women, 39.3% were not aware the vaccine could be given to those with a chronic disease, and 37.4% were unaware that those with suppressed immunity were eligible for vaccination. Moreover, 31.4% were unaware that the rabies vaccine could be given to adults aged >65 years (Table 5). When asked about the situations in which the rabies vaccination could be administered, 74.2–94.6% correctly knew that the vaccine should be administered in cases of bites or scratches from cats, dogs, or other animals with a risk of rabies. However, rates showed lower correct knowledge levels regarding contact other than bites and scratches (46.2–66.6%). The licking of healthy skin by a rabid or suspected rabid animal is a category 1 contact that does not require vaccination, but 49.2% of the respondents stated that rabies vaccination was required. The administration of prophylactic rabies vaccination was also included in the questionnaire. Although prophylactic rabies vaccination is not recommended for those who keep domestic pets, 76.8% of the students incorrectly recommended vaccination.

The students were further asked whether they had previously heard of rabies immunoglobin. The responses showed that 514 (57.6%) had and 378 (42.4%) had not heard of immunglobulin. When asked about the indications for rabies immunoglobulin, the rates of correct responses given for immunoglobulin were lower (between 15.9% and 57.8%) than those for vaccination (Table 5). Immunoglobulin administration is required in cases of Category 3 injuries, where the skin’s integrity is compromised. In total, only 57.8% of students correctly believed that immunoglobulin was necessary for a bleeding bite, and 51.6% for a bleeding scratch. While immunoglobulin application is not required for superficial category 2 injuries without compromised skin integrity, 84.1% of the students stated that immunoglobulin should be applied in cases featuring a superficial bite without bleeding, 66.3% stated that immunoglobulin administration is indicated for superficial scratching cases without bleeding.

### 4.6. Factors Associated with Rabies Knowledge

The responses provided by the students to the questions were collated, and the total rabies knowledge points and knowledge points among the subgroups were calculated for each student. The average score of all students regarding rabies knowledge was 41.82 ± 13.08. The maximum questionnaire score was 70, indicating that approximately 60% of the questions were answered correctly. Generally speaking, we found that as the year of study increased, the total and subgroup rabies knowledge points of the students also increased. A minimal decrease was observed only in the transition from the fifth to sixth year of study. We found a statistically significant difference in the rabies knowledge points of the students according to their year of study (*p* < 0.001 for all values) (Table 6).

Post hoc sub-group analyses were also performed based on the year of study. No significant differences were found between the first and third classes or between the fourth and fifth classes. Statistically significant differences were determined between each of the first, second, and third classes and between the fourth, fifth, and sixth classes. All students and those in their fourth, fifth, and sixth years of study were separated into two groups according to their previous experience with rabies lessons. The total and subgroup points of the students who had received rabies lessons were statistically significantly higher (*p* < 0.001 for all values). The students were also asked about their personal rabies knowledge. Among those who stated that they had knowledge of rabies, all points were found to be statistically significantly higher (*p* < 0.001 for all values). The students who reported that they or someone close to them (family member) had been bitten by an animal with suspected rabies presented statistically significantly higher total rabies knowledge, disease agent and characteristics, vaccination, and immunoglobulin points (*p* < 0.001 for all values), as well as higher animal transmission and clinical findings (*p* = 0.008) and wound care points (*p* = 0.004). Age and year of study were found to be correlated with rabies knowledge points. As the age and year of study increased, the rabies total and all subgroup points also increased.

## 5. Discussion

The rabies disease is endemic over a wide geographical area, primarily in Asian and African countries. As a disease that is fatal but preventable through vaccination, the administration of PEP and PrEP is life-saving [20]. Clinicians play a direct role in the determination and referral of individuals at risk. Therefore, a lack of knowledge about rabies will inevitably have a negative effect on mortality rates. In this context, the aim of this study was to investigate the knowledge levels of rabies among medical faculty students. The results of this study show that as the students’ year of study increases, so too does the level of rabies knowledge. In addition, those who had undertaken an infectious diseases internship or had a history of contact with suspected rabies (themselves or a family member) had a higher level of knowledge, but the students were observed to have a lower level of knowledge about the animals from which rabies is transmitted, as well as the administration of vaccines and immunoglobulin.

Rabies is a viral disease, which was correctly identified by 88% of the students in this study. Studies among medical students in India reported that the agent of this disease was correctly identified by 83.1% of students [16], 95% of interns and final-year medical students [17], 96.8% of third-year students [21], and 88.8% of first-year students [22]. In this study, students from all years (first to sixth year) were included as participants. The first-year students had not yet taken any courses on infectious diseases or microbiology, which could explain why 12% did not know that rabies is viral in origin. The management of cases with suspected risk of rabies contact starts with knowing the epidemiology of rabies. Understanding which animals can transmit rabies allows the correct management of patients presenting with contact or injury of animal origin. Dogs are most often responsible for the spread of rabies [3]. A previous study of medical students in India reported that 93.4% of the students identified dogs to be the most common reservoir of rabies, while 91.8% stated that the most common form of infection was through a bite from a rabid animal [16]. In another study, 98% of medical students identified dogs as the main carriers of rabies, and only 54% stated that other animals such as bats, cats, jackals, and pigs could be a source of rabies [17]. Other studies showed that medical students were less aware that infections could be caused by vectors other than a bite [14,16,22,23]. A study of first-year medical students in India reported that all students knew that rabies could be transmitted by dogs, 33% knew that it could be spread by cats, 16% knew it could be spread by wild animals, 3.3% stated that it could be transmitted with a scratch, and 1.1% stated that it could be transmitted through licking [22]. Rabies infection through a bite occurs via salivary transfer of the virus. In a study of final-year medical students, only 9% knew that saliva could lead to infection with rabies [23]. In a study of medical students in India, 91.8% stated that rabies could be spread from an animal bite, and 4.4% stated that infections could be caused through other vectors [16]. In the present study, most medical students knew that rabies is spread by dogs and cats (96.9% and 87.4%, respectively), whereas the status of rabies transmission from farm animals was incorrectly understood at a high rate (61.8–72.1%). In Türkiye, rabies is most commonly caused by dogs, followed by cats, livestock, and wild animals [2]. Therefore, in this study, we asked questions regarding the transmission of rabies originating from dogs, livestock, and wild animals. In Türkiye, free-roaming dogs and cats are frequently encountered on the streets, potentially leading to higher rates of attacks by such animals than those in countries where free-roaming animals are well managed. This factor may explain why dogs are known to be the most common rabies-transmitting animal among the participants. For animals with suspected rabies, a 10-day observation period is recommended only for dogs and cats [24]. In a study by Akçalı [14], 34.4% of final-year medical faculty students stated that a 10-day observation period was necessary for cats and dogs. Likewise, in a study by Twari [16], 24% of medical students stated that an observation period of at least 10 days was necessary. In the present study, a 10-day observation period was stated to be necessary for dogs by 88.6% of the medical faculty students and for cats by 73.5%. However, 59.8–79.1% stated that there should also be an observation period for farm animals and wild animals, which do not require this measure. In Türkiye, the number of rabies risk contact notifications totals around 250,000 per year. Most of the detected rabies cases are reported to be of dog origin (43.32%) [2]. In Türkiye, in cases of rabies risk contact, the affected individuals are also informed about the quarantine of the animal by the relevant authorities. Dogs were likely known to be quarantined more often among respondents due to the high numbers of reported rabies risk contact cases alongside the fact that rabies is mostly caused by dogs and the prevalence of free-roaming dogs in Türkiye. Many of the study participants (86.7–96.5%) stated that clinical findings, such as irritability, aggression, increased salivation, paralysis, and contractions could be observed among animals with suspected rabies. In a study in India, only 24.5% of students knew the symptoms of rabies in dogs [15]. Contact may be initially stopped with an animal that looks ill or aggressive when there is a suspicion of rabies; therefore, the clinical findings of rabies in animals should be known. Reducing contact with sick-looking animals and increasing preventive measures can help prevent the spread of rabies, especially in areas where free-roaming dogs and livestock are common.

In injuries with a risk of rabies, good wound care is an effective method for reducing viral infection. When there has been exposure, the wound area must be washed well with soap and plenty of water [2]. Giri et al. [21] found that most medical students (96.2%) were knowledgeable on the subject of immediately washing the wound with soap and water after an animal bite. Likewise, studies by Tiwari A. [16] and Praveen et al. [22], respectively, reported that 72.7% and 66.6% of medical students correctly knew that the wound should be immediately washed with soap and water. In wound care, washing with soap and water should be followed by cleaning with alcohol or iodine antiseptics [2]. However, previous studies have shown that the use of alcohol and antiseptics in wound care is less well known than washing with soap and water [14,16]. A previous study among university students reported that 64.5% of students were not aware of the importance of quickly washing bite wounds [12]. Another study evaluating patients presenting at Rabies Prevention Centers in eight Asian countries for post-exposure prophylaxis highlighted two important factors that could make a difference in the public being actively informed: the application of appropriate wound care and presenting at the nearest Rabies Prevention Centre as soon as possible [25]. The results of the current study show that while 78.9% of the medical students correctly understood that the wound should be cleaned with soap and plenty of water, 64.3% of respondents stated that the nearest health center should be approached to seek medical care prior to any wound intervention. Although the medical students in the current study were better educated than the general population, incorrect practices were accepted as correct at unacceptable rates, such as 34.8% for bleeding the wound and 24.4% for cleaning the wound with saliva. In our country, the management of patients with rabies risk contact is carried out in the emergency department. Students begin emergency department rotations in their sixth year and infectious disease internships during their fifth year. Consequently, students in years one to four may have less theoretical and practical knowledge about wound care than more advanced students. Rapid wound care is very important in preventing the transmission of rabies. Therefore, the fact that 64.3% of students believed they should seek medical attention without any initial wound intervention suggests a large knowledge gap in this area. This result indicates that such a lack of awareness is not dependent on year of study and highlights the need for improved education on proper wound care.

After wound care and treatment, the emergency management of rabies exposure involves vaccination in accordance with WHO standards, followed by the administration of rabies immunoglobulin to the wound if indicated [7]. However, the wound must first be categorized. In a study among sixth-year medical students, the vast majority categorized animal bite wounds correctly (Category 1—62%, Category 2—66%, and Category 3—72.0%) [23]. However, in another study, very few (11%, 5%, and 11%) interns and final-year medical students had a correct knowledge of wounds and Category 1 and 2 rabies [17]. In studies in India, 43.7% of medical students correctly understood the rabies vaccine schedule [16]; 16.6% of first-year students knew the vaccine doses [22]; and 50% and 29% of interns and final-year students, respectively, knew the vaccination schedule and vaccine doses in post-exposure prophylaxis [17]. A study in Türkiye reported that 29% of final-year medical students did not know the indications for rabies vaccine/rabies immunoglobulin administration [14]. The literature further demonstrates that medical students rarely know the indications for immunoglobulin [14,16,22,23]. In the present study, 74–94.6% of students correctly identified the need to administer the rabies vaccine in cases featuring contact with suspected Category 2 and 3 rabies, but the indications for immunoglobulin administration in cases with Category 2 and 3 contact were known at lower rates (15.9–57.8%). In the present study, 74–94.6% of students correctly recognized that rabies vaccination should be administered in cases involving biting or scratching by cats, dogs, and other animals suspected to have Category 2 and 3 rabies. Rabies cases in Turkey are mostly caused by dogs, and the number of free-roaming cats and dogs remains quite high. For these reasons, the participants may have been more aware of the need for vaccination in cases of cat- and dog-related bites and scratches. The licking of healthy skin by a rabid or suspected rabid animal is considered a Category 1 contact that does not require vaccination. However, 50.8% of the respondents stated that rabies vaccination was required in such cases. We observed lower correct knowledge levels among participants regarding contact vectors other than bites and scratches (46.2–66.6%). Unlike immunization and protection methods in other infectious diseases, rabies vaccination requires more detailed management, such as categorization of the contact. This factor may make it difficult for students to learn or remember the situations in which vaccination is required and may have decreased the level of such knowledge in this study. The students in the present study were also asked who is eligible for the rabies vaccine. In total, 45.2% of respondents did not know that the vaccine could be administered to pregnant women, 33.5% did not know infants that could be vaccinated, 31.4% did not know that the vaccine could be administered to individuals aged >65 years, 37.4% did not know that those with suppressed immunity could be vaccinated, and 39.3% did not know that those with a chronic disease could be vaccinated. In a study in India, 40.6% of medical students stated that pregnancy is not a contraindication for rabies vaccination [15]. In another study in Türkiye, only 29.8% of physicians stated that the vaccine could be safely administered to pregnant women [26]. However, it is stated in the relevant guidelines that pregnancy and lactation do not represent a contraindication for administration of the rabies vaccine and immunoglobulin; instead, the vaccine is effective and safe in such cases [7]. Since this study was conducted among medical students at all levels, students who did not receive clinical internships may not have known the indicated and contraindicated vaccines in certain risk groups, which may have decreased the mean knowledge level.

One vaccination strategy to prevent human rabies is immunity applied before encountering the risk factor—in other words, prophylaxis before contact. It was reported that 80% of interns and final-year students recommend vaccination among high-risk populations [17]. Other studies have shown that PrEP management is known by 17.9% of physicians in Türkiye [26] and 27% in France [27]. In this study, PrEP was considered to be necessary for veterinary practitioners by 91% of students, for those caring for animals by 90%, for those at high risk of contact with wildlife by 87.4%, and for those travelling to regions with a risk of rabid dogs by 82.4%.

A total of 70 parameters related to rabies were obtained via the questionnaire and scored. The mean points were determined to be 41.8. Since first-year students had not yet taken any classes related to infectious diseases, we ensured the questions were basic and did not request detailed information for each topic. On the other hand, the mean points for fifth- and sixth-year students were 49.9 and 49.4, respectively. As the year of study increased, the obtained points also increased statistically significantly. Specifically, the points of all students who had received rabies lessons and those among fourth-, fifth-, and sixth-year students who had undertaken clinical practice were significantly higher. Moreover, students who had a history of contact with suspected rabies either personally or in a close relative, as well as those who stated that they had information about rabies, had statistically significantly higher points. The need for medical care because of a dog attack may have led these individuals to access information about rabies, which may have increased their rabies knowledge. A study in India also reported that the levels of rabies knowledge increased together with an increase in the years of study of medical students [15]. In another study among university students, the level of rabies knowledge was overall moderate, except among those studying health sciences, who had significantly higher scores. Those who owned a dog or had someone close to them who had been bitten by a dog also had a significantly higher level of knowledge [12]. Negative experiences, like being bitten by a dog, may have facilitated the retention of information about rabies.

Previous studies evaluating the level of rabies knowledge among medical students have reported insufficient levels of rabies knowledge [14,15,16,17]. The importance of education is indisputable on this point. It has been shown that even one hour of rabies training given to third-year medical students statistically significantly increased their knowledge on epidemiology and prophylaxis [18]. Moreover, concept mapping was found to be more effective for sixth-year students than traditional textbook reading in learning rabies surveillance [28].

## 6. Limitations

Despite its practical contributions, this study also has some limitations. First, the study was cross-sectional and self-reported in design, which could have led to recall bias. Although we sought to measure knowledge levels with detailed questions created by the researchers using the literature, there remains no validated questionnaire on this subject. Moreover, since this was a single-center study, the results cannot be generalized to the whole country or other medical faculties. Since first-grade students did not receive any education about the rabies vaccine and disease, in order to address all grades, questions related to the vaccination schedule and vaccine types were not asked in detail. Lastly, this study measured only the level of knowledge. Thus, there remains a need for studies designed to also measure attitudes and behaviors.

## 7. Conclusions

The results of this study indicate that the level of rabies knowledge among medical students is insufficient. Considering that rabies is a fatal disease that can be prevented through vaccination, it is of vital importance to address relevant gaps in knowledge and organize educational efforts to correct misinformation for patients presenting with a suspected risk of rabies. Identifying incorrect points and determining the level of knowledge on this subject among students would be of great value for a fatal disease, such as rabies. Several efforts could ensure a more adequate level of rabies knowledge, including increasing the duration of training related to rabies prior to graduation, using methods that have been shown to be more effective (e.g., concept maps), measuring the level of knowledge among students, and planning training to cover missing or incorrectly understood concepts.

## Figures and Tables

**Table 1 tropicalmed-10-00009-t001:** Genders of the medical students, characteristics according to year of study, and ownership of domestic pets.

		*n* *	%
Gender	Female	452	50.7
	Male	440	49.3
	Total	892	100
Year of study	1st year	158	17.7
	2nd year	146	16.4
	3rd year	103	11.5
	4th year	138	15.5
	5th year	191	21.4
	6th year	156	17.5
	Total	892	100
Having a domestic pet	Yes	97	10.9
	No	795	89.1
	Total	892	100
Routine vaccination of domestic pet	Yes	65	56.9
	No	28	43.1
	Total	93	100
History of being bitten by domestic pet	Yes	59	61
	No	36	49
	Total	95	100
History of being bitten by an animal with suspected rabies (self or family member)	Yes	137	15.4
	No	755	84.6
	Total	892	100
Status of having received lessons about rabies (for all classes)	Yes	254	29.1
	No	618	70.87
	Total	872	100
Status of having received lessons about rabies (for 4th-, 5th-, and 6th-year students)	Yes	254	51.8
	No	211	44.2
	Total	465	100
Do you have knowledge about rabies disease?	Yes	676	77.1
	No	200	22.8
	Total	876	100

* The number of *n* varies here due to incomplete responses.

**Table 2 tropicalmed-10-00009-t002:** The questions and responses related to animals transmitting rabies and the observation periods.

	Which Animal or Animals Are at Risk of Rabies Transmission After Suspected Contact ? * (*n* = 892)				When There Is Suspected Rabies Contact, for Which Animals Should There Be a 10-Day Observation Period (Quarantine)? ** (*n* = 892)			
	Correct Response		Incorrect Response		Correct Response		Incorrect Response	
	*n*	%	*n*	%	*n*	%	*n*	%
Dogs ***^,^****	864	96.9	28	3.1	790	88.6	102	11.4
Cats *^,^**	780	87.4	112	12.6	656	73.5	236	26.5
Wolves *	601	67.4	291	32.6	186	20.9	706	79.1
Foxes *	559	62.7	333	37.3	204	22.9	688	77.1
Snakes	410	46.0	482	54.0	359	40.2	533	59.8
Winged farm animals	341	38.2	551	61.8	332	37.2	560	62.8
Rabbits	326	36.5	566	63.5	335	37.6	577	62.4
Horses *	318	35.7	574	64.3	291	32.6	601	67.4
Donkeys *	310	34.8	582	65.2	298	33.4	594	66.6
Mice/rats	291	32.6	601	67.4	303	34	589	66
Cows *	278	31.2	614	68.8	310	34.8	582	65.2
Goats *	252	28.3	640	71.7	323	36.2	569	63.8
Sheep *	249	27.9	643	72.1	315	35.3	577	64.7

The correct response for questions marked with * or ** was “yes”; otherwise, the correct response was “no”.

**Table 3 tropicalmed-10-00009-t003:** Questions and responses related to wound care.

	Correct Response		Incorrect Response	
	*n*	%	*n*	%
The wound area should first be washed with plenty of water and soap. *	704	78.9	188	21.1
The wound should first be cleaned with cologne, alcohol, or one of the iodine antiseptics.	379	42.5	513	57.5
Blood should be drawn from the wound.	582	65.2	310	34.8
The wound should be cleaned with saliva.	674	75.6	218	24.4
The person should be taken to the nearest healthcare facility without any wound intervention.	318	35.7	574	64.3

The correct response for questions marked with * was “yes”; otherwise, the correct response was “no”.

**Table 4 tropicalmed-10-00009-t004:** Questions and responses related to the rabies agent and disease characteristics.

		Correct Response		Incorrect Response	
		*n*	%	*n*	%
What is the causative agent of rabies disease?	Virus *	785	88	107	12
What kind of disease is rabies?	It is a treatable disease	313	35.1	579	64.9
	The disease is only present in animals	796	89.2	96	10.8
	It is a very serious fatal disease *	769	86.2	123	13.8
	It is a contagious disease *	748	83.9	144	16.1
	It is a disease that can be prevented with vaccination *	813	91.1	79	8.9
	It is a severe but not fatal disease	686	76.9	206	23.1

The correct response for questions marked with * was “yes”; otherwise, the correct response was “no”.

**Table 5 tropicalmed-10-00009-t005:** Questions and responses related to the vaccine and immunoglobulin administration.

		Correct Response		Incorrect Response	
		*n*	%	*n*	%
Who can be vaccinated against rabies?	Children *	773	86.7	119	13.3
	Older adults aged >65 years *	612	68.6	280	31.4
	Infants *	593	66.5	299	33.5
	Individuals with low immunity *	558	62.6	334	37.4
	Individuals with chronic disease *	541	60.7	351	39.3
	Pregnant women *	489	54.8	403	45.2
In what situations should rabies vaccine be given?	Scratch from a stray cat *	662	74.2	230	25.8
	Bite from a stray cat *	783	87.8	109	12.2
	Scratch from a stray dog *	702	78.7	190	21.3
	Bite from a stray dog *	844	94.6	48	5.4
	Bite from an animal with rabies or suspected rabies *	840	94.2	52	5.8
	Scratch from an animal with rabies or suspected rabies *	747	83.7	145	16.3
	Slaughtering an animal with rabies or suspected rabies without protective equipment (gloves, goggles, etc.) *	594	66.6	298	33.4
	Healthy skin being licked by an animal with rabies or suspected rabies	453	50.8	439	49.2
	Contact between healthy skin and the blood of an animal with rabies or suspected rabies	432	48.4	460	51.6
	Eating the cooked meat of an animal with rabies or suspected rabies	412	46.2	480	53.8
	Drinking the boiled milk of an animal rabies or with suspected rabies	412	46.2	480	53.8
To whom should the rabies vaccine be given without contact with rabies infected animal?	Veterinary practitioners *	812	91.0	80	9.0
	Personnel working in animal shelters *	810	90.8	82	9.2
	People looking after animals *	803	90.0	89	10.0
	People engaged in outdoor sports with a high risk of contact with wildlife or those spending time with wild animals *	780	87.4	112	12.6
	People with domestic pets	207	23.2	685	76.8
	People travelling to countries with a high rate of rabies among dogs *	735	82.4	157	17.6
Indication for rabies immunoglobulin	Superficial scratch with no bleeding	301	33.7	591	66.3
	Scratch with bleeding *	460	51.6	432	48.4
	Superficial bite with no bleeding	142	15.9	750	84.1
	Bite with bleeding *	516	57.8	376	42.2

The correct response for questions marked with * was “yes”; otherwise, the correct response was “no”.

**Table 6 tropicalmed-10-00009-t006:** Relationships between gender, ownership of pets, and year of study among students with total rabies knowledge points and subgroup points.

		TRKP ^1^ (Mean ± SD)	*p*	DCCAP ^2^ (Mean ± SD)	*p*	ATCSP ^3^ (Mean ± SD)	*p*	WCP ^4^ (Mean ± SD)	*p*	VIP ^5^ (Mean ± SD)	*p*
Class	Class 1	32.40 ± 12.90	<0.001	4.89 ± 1.48	<0.001	12.14 ± 6.85	<0.001	2.21 ± 1.30	<0.001	13.16 ± 5.91	<0.001
	Class 2	33.98 ± 10.01		4.94 ± 1.55		12.42 ± 5.80		1.75 ± 1.34		14.87 ± 4.72	
	Class 3	36.50 ± 10.88		5.31 ± 1.29		12.55 ± 6.14		2.61 ± 1.39		16.03 ± 5.11	
	Class 4	44.93 ± 10.39		5.53 ± 1.10		17.03 ± 7.07		3.39 ± 1.41		18.99 ± 4	
	Class 5	49.96 ± 11.37		6.10 ± 0.98		18.70 ± 7.71		3.72 ± 1.19		21.43 ± 4.20	
	Class 6	49.48 ± 8.50		6.03 ± 1.06		17.79 ± 6.00		3.88 ± 1.07		21.78 ± 3.08	
	Total	41.82 ± 13.08		5.5 ± 1.34		15.38 ± 7.26		2.98 ± 1.51		17.95 ± 5.66	
Rabies Course Taking Status (years 4, 5, and 6)	Yes	51.88 ± 9.35	<0.001	6.11 ± 1.03	<0.001	19.53 ± 6.74	<0.001	3.93 ± 1.05	<0.001	22.31 ± 3.09	<0.001
	No	44.22 ± 9.99		5.71 ± 1.06		16.00 ± 6.87		3.44 ± 1.35		19.08 ± 4.18	
Rabies Course Taking Status (for all classes)	Yes	51.88 ± 9.35	<0.001	6.11 ± 1.03	<0.001	19.53 ± 6.74	<0.001	3.93 ± 1.05	<0.001	22.31 ± 3.09	<0.001
	No	37.49 ± 12.03		5.25 ± 1.38		13.59 ± 6.72		2.59 ± 1.50		16.06 ± 5.48	
Gender	Female	41.49 ± 13.49	0.452	5.47 ± 1.38	0.398	15.27 ± 7.47	0.636	2.97 ± 1.50	0.812	17.59 ± 5.84	0.387
	Male	42.15 ± 12.64		5.54 ± 1.31		15.50 ± 7.04		2.99 ± 1.52		18.12 ± 5.47	
Presence of Pet Animals	Yes	44.28 ± 13.07	0.05	5.7 ± 1.18	0.128	16.77 ± 7.42	0.052	3.01 ± 1.45	0.828	18.79 ± 5.40	0.121
	No	41.52 ± 13.05		5.48 ± 1.36		15.21 ± 7.22		2.97 ± 1.52		17.85 ± 5.68	
Vaccination status of Pet	Yes	44.69 ± 11.87	0.937	5.66 ± 1.14	0.370	16.95 ± 6.54	0.925	3.11 ± 1.43	0.583	18.97 ± 5.17	0.722
	No	44.46 ± 14.38		5.89 ± 1.10		17.11 ± 8.51		2.93 ± 1.43		18.54 ± 5.84	
History of Bite by an Animal with Suspected Rabies	Yes	45.71 ± 12.31	<0.001	5.87 ± 1.14	<0.001	16.85 ± 6.92	0.008	3.31 ± 1.45	0.004	19.68 ± 5.61	<0.001
	No	41.11 ± 13.09		5.44 ± 1.37		15.12 ± 7.29		2.92 ± 1.51		17.64 ± 5.61	
Knowledge about Rabies Disease	Yes	44.74 ± 11.82	<0.001	5.77 ± 1.15	<0.001	16.53 ± 7.02	<0.001	3.25 ± 1.42	<0.001	19.19 ± 5.05	<0.001
	No	31.88 ± 12.25		4.62 ± 1.56		11.39 ± 6.54		2.03 ± 1.40		13.84 ± 5.73	

^1^ TRKP: Total Rabies Knowledge Points; ^2^ DCCAP: Disease–Clinical and Causative Agent Points; ^3^ ATCSP: Animal Transmission and Clinical Signs Points; ^4^ WCPs: Wound Care Points; ^5^ Vaccine and Immunoglobulin Points.

## Data Availability

Research data available upon request by contacting the lead author.

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
