# Peer review of "Rabies Disease and Prophylaxis Knowledge Among Turkish Medical Students: Insights from a Cross-Sectional Study"

_tropicalmed, 2024, doi:10.3390/tropicalmed10010009_

Round 1

Reviewer 1 Report

Comments and Suggestions for Authors

The authors have described a questionnaire to evaluate the knowledge of various classes of medical students in a university. The article is well written in all aspects; however, certain parameters need to be addressed by the authors to qualify the quality and rigor of the manuscript;

Major comments;

1. The authors did not disclose the language of the questionnaire, whether it was in Turkish or English, or another type of language, moreover, the whole questionnaire needs to be deposited in the supplementary data to replicate the study for the authors.

2. The article contains various parameters or questions; a few are related to attitude and practice (for example, the administration of immunoglobulins). The authors may employ typical questions related to knowledge, attitude, and practice to evaluate students' perceptions of rabies.

3. The totality or total responses at some tables (I highlighted in red) are different, as it seems some of the responses were missing.

4. The authors did not give due attention to the results as it seems that all the medical students do not have the appropriate or required knowledge of rabies. The topic of rabies is important to medical students, but even then, the levels of the variables are variable. How authors would address this concern in the future to revise the syllabus of the students. What kind of measures or recommendations authors have planned out of this study?

5. The timeline to conduct the study was given in the manuscript, which means that a few months were required to complete the survey under stratified random sampling? How authors justify this much period? Was the proforma delivered to students in hard form or soft form?

6. It is important to mention the brief epidemiology of rabies variants in Turkey or to talk about which variant of rabies is prevalent in Turkey and which type of rabies the authors are talking about? it is dog-mediated rabies or wild rabies? Questions regarding the schedule of vaccines and types of variants were missing which are important. Moreover, no questions were inserted regarding one health approach or zoonotic importance.

7. Authors need to show the data analysis and data evaluation in the form of excel or worksheets in supplementary files

Comments on the Quality of English Language

1. The whole manuscript is required to be screened for the English language by a native English speaker to improve the readability, comprehension, and understanding of the manuscript.

2. The English language needs attention specifically in certain paragraphs needs revision of completely English statements to rectify grammar mistakes. The questions in the proforma for students also need revision.

Author Response

For research article

‘Rabies Disease and Prophylaxis Knowledge among Medical Students: Insights from a Cross-Sectional Study’

Response to Reviewer X Comments

Dear Reviewer,

Thank you very much for taking the time to review this manuscript.

I have carefully reviewed your valuable feedback and have made the necessary revisions to the manuscript in line with your suggestions. Please find the detailed responses below and the corresponding revisions/corrections highlighted/in track changes in the re-submitted files.

I truly appreciate the time and effort you have dedicated to providing these insightful comments, which have significantly contributed to improving the quality of the manuscript. Should you have any additional suggestions or revisions, I would be more than happy to address them.

Thank you once again for your guidance and support throughout this process.

Kind regards,

Point-by-point response to Comments and Suggestions for Authors

REVIEWER 1

Comment 1: The authors did not disclose the language of the questionnaire, whether it was in Turkish or English, or another type of language, moreover, the whole questionnaire needs to be deposited in the supplementary data to replicate the study for the authors

Response 1: Thank you for pointing this out. We agree with this comment. Therefore, we have added to the material method the fact that the questionnaire is prepared and administered in Turkish.  We have provided both the questions and their answers in the findings section to ensure all questions in the questionnaire are included. Therefore, we believe adding the entire questionnaire as a supplementary file may not be necessary, as it could contribute to unnecessary document load. However, it is important to emphasize that this is a questionnaire, not a scale. Researchers are encouraged to adapt questions to the specific characteristics of the regions they study. Of course, they are also welcome to incorporate the questions from our survey into their research.

Comments 2: The article contains various parameters or questions; a few are related to attitude and practice (for example, the administration of immunoglobulins). The authors may employ typical questions related to knowledge, attitude, and practice to evaluate students' perceptions of rabies:

Response 2: Since the participants were students and had no experience or authorization to administer vaccines or immunoglobulins, only the level of knowledge about rabies disease, transmission and vaccine immunoglobulin administration practice was questioned. In order to avoid confusion both when collecting the data and writing the findings, sections such as contamination, wound care, vaccination immunoglobulin application etc. were grouped together, asked in this way and added as findings.

Comment 3: The totality or total responses at some tables (I highlighted in red) are different, as it seems some of the responses were missing.

Response 3: Yes, the number of ‘n’ varies due to incomplete responses. Not all questions were completed by the students, some were left blank. Therefore the total number 'n' is given in the row for each descriptive data and differed from each other. An explanation has been added below the table 1 to avoid confusion.

Comment 4: The authors did not give due attention to the results as it seems that all the medical students do not have the appropriate or required knowledge of rabies. The topic of rabies is important to medical students, but even then, the levels of the variables are variable. How authors would address this concern in the future to revise the syllabus of the students. What kind of measures or recommendations authors have planned out of this study?

Response 4: Suggestions for increasing the level of knowledge have been added at the end of the conclusion section

Comment 5: The timeline to conduct the study was given in the manuscript, which means that a few months were required to complete the survey under stratified random sampling? How authors justify this much period? Was the proforma delivered to students in hard form or soft form?

Response 5: The students were working in different clinics and hospitals and taking student courses in different lecture theatres at different times. Therefore, it took time to reach the students, the reached students were invited to the study and participation was voluntary.

Comment 6: It is important to mention the brief epidemiology of rabies variants in Turkey or to talk about which variant of rabies is prevalent in Turkey and which type of rabies the authors are talking about? it is dog-mediated rabies or wild rabies? Questions regarding the schedule of vaccines and types of variants were missing which are important. Moreover, no questions were inserted regarding one health approach or zoonotic importance.-epidemiology of Turkey mentioned , type of rabies mentioned dog- mediated.

Response 6: Which rabies species is common in Turkey and which rabies species we are talking about is added to the relevant place in the discussion section. Since class 1 students have not received any training on rabies vaccination and disease and in order for the questionnaire to address all classes, the vaccination schedule and vaccine types were not asked. This situation is also mentioned in the limitation section. The section of the questionnaire containing information contained 70 items and the section in which descriptive characteristics were questioned contained 10 questions. Since the increase in the number of questions may reduce the voluntariness to participate in the research, the attention of the participants may be distracted as the questionnaire is filled out longer, and the rate of leaving empty answers may increase, the questionnaire form was limited while preparing the questionnaire form. Since this study has been completed, it will not be possible to add questions related to a health approach or zoonotic importance. I will plan to add questions related to a health approach or zoonotic importance in other studies I will conduct on the level of knowledge about rabies, thank you for your contribution.

Comment 7: Authors need to show the data analysis and data evaluation in the form of excel or worksheets in supplementary files:

Response 7: Our ethics committee commission does not favour sharing raw data with 3rd parties. In addition, in the documents submitted to the ethics committee commission and in the informed consent form, it was promised that the data would not be shared with third parties. Permission can be obtained from the ethics committee commission within justified reasons. 

Response 8: The sections highlighted in red in the review PDF have been thoroughly reviewed, and the necessary corrections have been made in the relevant parts of the article, along with the addition of supporting evidence in the PDF file.

Response 9: I am sorry for the negative feedback about the English language of the article, I had the support of a professional translator native English speaker to avoid this situation. We asked for help from the MDPI English editing service in response to a proposal to improve the English language and I will be uploading an edited version of the article.

Additional clarifications

All changes and additions are phosphorised (yellow) and marked at the attached file.

Reviewer 2 Report

Comments and Suggestions for Authors

The article is written in such a poor English that it prevents my understanding of what is going on. After the article is rewritten, it should be reviewed again.

Overall, the topic and content is likely good enough for the publication but there are still some issues.

For example, rabies can be treated if caught soon enough.

I am not sure what is the authors really discussing in the section "Discussion". I believe they should be discussing how they results compare to other studies but they seem to be discussing facts about the disease. While facts about the disease are perhaps important part of the article, stating the facts in the intro or in connection with the correct answers to survey questions would be more appropriate.

Some facts may have a more local character. For example, in Americas, bats and other animals play a crucial role in the transmission. 

Comments on the Quality of English Language

The quality of English is so low that it prevents proper understanding. The very first "sentence" of the abstract is not a sentence and issues like that are too numerous to list

Author Response

For research article

‘Rabies Disease and Prophylaxis Knowledge among Medical Students: Insights from a Cross-Sectional Study’

Response to Reviewer X Comments

Dear Reviewer,

Thank you very much for taking the time to review this manuscript.

I have carefully reviewed your valuable feedback and have made the necessary revisions to the manuscript in line with your suggestions. Please find the detailed responses below and the corresponding revisions/corrections highlighted/in track changes in the re-submitted files.

I truly appreciate the time and effort you have dedicated to providing these insightful comments, which have significantly contributed to improving the quality of the manuscript. If there are still areas that require further improvement, I would be more than willing to make additional changes to enhance the quality of the manuscript. Your guidance is greatly appreciated, and I thank you again for helping refine this work.

Thank you once again for your guidance and support throughout this process.

Kind regards,

Point-by-point response to Comments and Suggestions for Authors

Comment: The article is written in such a poor English that it prevents my understanding of what is going on. After the article is rewritten, it should be reviewed again.

The quality of English is so low that it prevents proper understanding. The very first "sentence" of the abstract is not a sentence and issues like that are too numerous to list

Response: I am sorry for the negative feedback about the English language of the article, I had the support of a professional translator native English speaker to avoid this situation. We asked for help from the MDPI English editing service in response to a proposal to improve the English language and I will be uploading an edited version of the article.

Comment: Overall, the topic and content is likely good enough for the publication but there are still some issues.

Response: I have a special interest in rabies and I have research in other special groups in which I have evaluated the level of knowledge and completed the data and waiting to be written. your positive feedback on the subject is a pleasure for me. your contributions to this article as feedback will surely have positive effects on future research. I am ready to do my best to make this article better if you have additional feedback for this article or where you do not consider it sufficient.

Comment: For example, rabies can be treated if caught soon enough. I am not sure what is the authors really discussing in the section "Discussion". I believe they should be discussing how they results compare to other studies but they seem to be discussing facts about the disease. While facts about the disease are perhaps important part of the article, stating the facts in the intro or in connection with the correct answers to survey questions would be more appropriate. Some facts may have a more local character. For example, in Americas, bats and other animals play a crucial role in the transmission. 

Response: I revised article due to feedback and made additions and subtractions. All changes and additions are phosphorised and marked in the article in blue at the attached file.

Round 2

Reviewer 1 Report

Comments and Suggestions for Authors

Thank you for making revisions to the manuscript. I still suggest to revise the title of the manuscript. regards

Author Response

Dear Reviewer,

Thank you for your feedback. I am pleased to inform you that we have carefully revised the manuscript in accordance with the suggestions. Specifically:

  1. Title Amendment: As suggested by the reviewer, the title has been amended to:
    "Rabies Disease and Prophylaxis Knowledge among Turkish Medical Students: Insights from a Cross-Sectional Study."
  2. Checklist Compliance:
  • All references have been reviewed to ensure relevance to the manuscript content.
  • Revisions made to the manuscript have been highlighted for easy identification by the editors and reviewers.
  • A detailed cover letter responding to the reviewers’ comments and explaining the revisions has been included.
  • No references were recommended by the reviewers, so no additional references were included.

If there are any further questions or clarifications required, please do not hesitate to contact me.

Thank you for your time and consideration.

Best regards,

Vasfiye DEMİR PERVANE

Assistant Professor, Dicle University Faculty o Medicine, Family Medicine Department, Diyarbakır, Türkiye

vasfiyyedemir@hotmail.com, 00905325158900

Reviewer 2 Report

Comments and Suggestions for Authors

The article improved significantly and it is now acceptable for the publication

Author Response

(The authors gave the same response as above.)
